# Elevated Serum Levels of YKL-40, YKL-39, and SI-CLP in Patients with Treatment Failure to DMARDs in Patients with Rheumatoid Arthritis

**DOI:** 10.3390/biomedicines12071406

**Published:** 2024-06-25

**Authors:** José David Tadeo Esparza-Díaz, Jorge Ivan Gamez-Nava, Laura Gonzalez-Lopez, Ana Miriam Saldaña-Cruz, Andrea Carolina Machado-Sulbaran, Alberto Beltrán-Ramírez, Miryam Rosario Guillén-Medina, Ana Gabriela Flores-Vargas, Edsaúl Emilio Pérez-Guerrero

**Affiliations:** 1Doctorado en Farmacología, Centro Universitario de Ciencias de la Salud, Universidad de Guadalajara, Guadalajara 44340, Jalisco, Mexico; davidtadeo.esparza@alumnos.udg.mx (J.D.T.E.-D.); ivangamezacademicoudg@gmail.com (J.I.G.-N.); lauraacademicoudg@gmail.com (L.G.-L.); ana.saldanac@academicos.udg.mx (A.M.S.-C.); rosario.guillen@alumnos.udg.mx (M.R.G.-M.); ana.flores3590@alumnos.udg.mx (A.G.F.-V.); 2Instituto de Investigación en Ciencias Biomédicas, Centro Universitario de Ciencias de la Salud, Universidad de Guadalajara, Guadalajara 44340, Jalisco, Mexico; dralbertobeltran@gmail.com; 3Departamento de Fisiología, Centro Universitario de Ciencias de la Salud, Universidad de Guadalajara, Guadalajara 44340, Jalisco, Mexico; 4Instituto de Investigación en Cáncer en la Infancia y Adolescencia, Centro Universitario de Ciencias de la Salud, Universidad de Guadalajara, Guadalajara 44340, Jalisco, Mexico; andrea.machado5223@academicos.udg.mx

**Keywords:** rheumatoid arthritis, treatment failure, chitinase-like proteins, YKL-40, YKL-39, SI-CLP

## Abstract

Around 30–60% of patients with rheumatoid arthritis (RA) present treatment failure to conventional synthetic disease-modifying antirheumatic drugs (csDMARDs). Chitinase-like proteins (CLPs) (YKL-40, YKL-39, SI-CLP) might play a role, as they are associated with the inflammatory process. This study aimed to evaluate CLP utility as a biomarker in the treatment failure of csDMARDs. A case–control study included 175 RA patients classified into two groups based on therapeutic response according to DAS28-ESR: responders (DAS28 < 3.2); non-responders (DAS28 ≥ 3.2). CLP serum levels were determined by ELISA. Multivariable logistic regression and receiver operating characteristic (ROC) curves were used to evaluate CLPs’ utility as biomarkers of treatment failure. Non-responders presented higher levels of YKL-40, YKL-39, and SI-CLP compared with responders (all: *p* < 0.001). YKL-40 correlated positively with YKL-39 (rho = 0.39, *p* < 0.001) and SI-CLP (rho = 0.23, *p* = 0.011) and YKL-39 with SI-CLP (rho = 0.34, *p* < 0.001). The addition of CLPs to the regression models improves diagnostic accuracy (AUC 0.918) compared to models including only clinical classical variables (AUC 0.806) *p* < 0.001. Non-responders were positive for all CLPs in 35.86%. Conclusions: CLPs could be considered as a useful biomarker to assess treatment failure, due to their association with clinical variables and improvement to the performance of regression models.

## 1. Introduction

Rheumatoid arthritis (RA) is a chronic autoimmune disease of unknown etiology that involves genetic, environmental, immunologic, and hormonal factors [1,2]. The primary goal of RA treatment is to achieve and maintain disease remission or disease activity levels as low as possible [3]. Disease-modifying antirheumatic drugs (DMARDs) have been used to achieve these objectives [1]. In Latin American countries, pharmacological treatment is mainly based on conventional synthetic DMARDs (csDMARDs) [4,5,6]. However, despite significant improvements in the majority of RA patients with csDMARDs, a notable proportion (30–60%) experienced treatment failure with csDMARDs [7,8,9]. Therefore, it is crucial to have tools that help identify patients with early treatment failure in order to prevent irreversible structural damage, disability, decreased quality of life, and persistent inflammatory states [10,11,12]. Although traditionally used as treatment failure biomarkers, rheumatoid factor (RF), anti-citrullinated peptide antibodies (ACPAs), and anti-peptidyl arginine deiminase 4 antibodies (anti-PAD4) have not proven to be useful in identifying patients with DMARD treatment failure [13,14,15]. In this regard, chitinase-like proteins (CLPs) might play a role, as they are associated with the inflammatory process in RA patients [16]. CLPs are proteins without enzymatic activity that lack six cysteine residues, resulting in conformational changes in their active site [17]. Three main CLPs are known in humans: YKL-40 (CHI3L1: chitinase-3-like-1), YKL-39 (CHI3L2: chitinase-3-like-2), and SI-CLP: Stabilin-interacting chitinase-like protein (CHID1: Chitinase domain-containing protein 1) [18]. YKL-40 has been primarily studied in RA and is associated with increased clinical disease activity and higher serum levels in patients with high disease activity [19,20]. To the best of our knowledge, YKL-39 and SI-CLP have not been extensively studied in RA patients. However, information from other diseases suggests that both YKL-39 and SI-CLP play crucial roles in inflammation and joint damage [21,22].

YKL-40, YKL-39, and SI-CLP present a structural relationship between them [16,23,24,25]. However, the in vivo relationships among these three structurally related CLPs are not well understood. Furthermore, the role of these three CLPs in RA and treatment failure based on csDMARDs is not known. Therefore, this study aimed to evaluate the utility of CLPs as biomarkers to identify RA patients with treatment failure.

## 2. Materials and Methods

### 2.1. Study Design

Case–control study. The study included 175 RA patients classified according to the 1987 ACR criteria [26] who voluntarily agreed to participate. The inclusion criteria were as follows: (a) females, (b) over 18 years old, and (c) pharmacological treatment based on csDMARDs. Exclusion criteria included: (a) overlap syndrome; (b) use of vitamins, antioxidants, and dietary supplements known to affect YKL-40 production [19]; (c) pregnant or lactating women; (d) treatment with biologic DMARDs. Patients with RA were recruited from the outpatient clinic of the Research Department of the University Center of Health Sciences, University of Guadalajara.

### 2.2. Clinical Assessments

All patients were evaluated by a rheumatologist using a structured questionnaire to collect sociodemographic variables, personal pathological history, and disease activity. Disease activity was assessed using the Disease Activity Score 28 combined with erythrocyte sedimentation rate (DAS28) [27]. Sedentary behavior was considered as any waking behavior, such as sitting, lying down, or reclining, characterized by an energy expenditure of 1.5 METs (metabolic equivalent of task) or lower, according to the definition of the Sedentary Behavior Research Network and World Health Organization [28].

### 2.3. Study Groups

Patients were divided into two groups based on their therapeutic response according to the DAS28-ESR: responders to csDMARDs and non-responders to csDMARDs. The responder group included patients with RA who met one of the following criteria: (1) a DAS28 score of less than 3.2 for at least six months prior to inclusion in the study without any changes to their csDMARD pharmacological regimen; or (2) a DAS28 score of 3.2 or greater who achieved a 50% reduction in their DAS28 score over a six-month period without any changes to their csDMARD pharmacological regimen. The non-responder group comprised patients with RA who met one or both of the following criteria: (1) active patients with a DAS28 score of 3.2 or greater who did not achieve a 50% reduction in their DAS28 score over the six-month period prior to study inclusion, despite treatment with csDMARDs; or (2) active patients with a DAS28 score of 3.2 or greater who remained active and required the addition of another csDMARD, as determined by the treating physician, within the six-month period prior to study inclusion. Appendix A shows the conformation of responders and non-responders, based on previous clinical characteristics.

### 2.4. Determination of Serum Levels of Chitinase-like Proteins (YKL-40, YKL-39, and SI-CLP)

Fasting blood samples were collected, and serum was stored at −80 °C until laboratory analysis. ELISA kits from MyBioSource, San Diego, CA, USA, with catalog numbers MBS765293 (YKL-40), MBS450589 (YKL-39), and MBS764700 (SI-CLP) were used to determine CLP serum levels. All samples were analyzed in duplicate by investigators blinded to the patient’s clinical characteristics, following the manufacturer’s instructions.

### 2.5. Other Laboratory Parameters

Additional samples were obtained for ESR (mm/h), C-reactive protein (CRP; mg/dL), rheumatoid factor (RF; IU/mL), and ACPAs (UI/mL). These determinations were performed according to the manufacturer’s instructions by investigators who were blinded to the patient’s clinical characteristics.

### 2.6. Statistical Analysis

Data normality for the main study variables (CLPs) was assessed using the Kolmogorov–Smirnov–Lilliefors test and graphs, indicating a non-normal distribution. Quantitative variables were expressed as median and interquartile range (IQR), and qualitative variables were expressed as frequencies and percentages (%). The Mann–Whitney U test was used for the comparison of quantitative variables between two independent groups, and the chi-square test (or Fisher’s exact test for small expected counts) for the comparison of proportions. Spearman’s correlation test was used to identify the strength of correlation between serum levels of CLPs and other quantitative clinical variables. To understand the influence of different variables on CLPs at various points in their distribution, quantile regression was performed, one for each CLP. These regressions were adjusted for variables associated with CLPs in the bivariate correlation analysis, as well as for variables with biological plausibility. In these regressions, it was validated that the models were adequate by verifying the independence and normality of the residuals and the linearity of the relationships between the independent and dependent variables in each analyzed quantile. Additionally, multicollinearity between the variables included in the models was examined. Receiver operating characteristic (ROC) curves were used to evaluate the CLP concentrations that best identified RA patients with treatment failure. The Youden index was used to select the optimal cut-off point. The area under the curve (AUC), 95% confidence intervals, sensitivity, specificity, positive predictive value (PPV), negative predictive value (NPV), and likelihood ratios for these serum levels were calculated to assess their utility in identifying RA patients with csDMARD treatment failure (non-responders). Logistic regression models were used to adjust for associated and confounding variables related to treatment failure. Initially, a model that included only clinical variables, excluding CLPs, was selected. Variables evaluated in this model were those with a *p*-value < 0.20 in univariate analysis, and biologically plausible. The final reference model was chosen considering Akaike information criterion (AIC) and Bayesian Information criterion (BIC) values. Additionally, the normality of the residual error and multicollinearity were used to validate the logistic regression models. Subsequently, to assess whether the addition of chitinases improved the performance of the reference model in identifying treatment failure, 14 new models were created, including each CLP and all possible combinations. ROC curves for each model were compared using the non-parametric DeLong test [29]. For all statistical analyses, the R programming language, R Core Team, Viena, Austria, version 4.3 was used, employing the following R packages: tidyverse v2.0.0; [30], ggcorrplot v0.1.4.1; [31], ggpubr v0.6.0; [32], rstatix v0.7.2; [33], pROC v1.18.4; [34], cutpointr v1.1.2; [35], epiR v2.0.65; [36], ggalluvial v0.12.5; [37] and quantreg v5.97; [38]. Statistical significance was set at *p* < 0.05.

## 3. Results

### 3.1. Comparison of Clinical Characteristics of RA Responders and Non-Responders

A total of 175 patients with RA were included: 87 in the responder group and 88 in the non-responder group. Table 1 describes and compares the clinical and sociodemographic characteristics of the two RA patient groups (responders and non-responders). The groups were similar in terms of sex, age, tobacco use, comorbidities, and rheumatoid factor levels. Additionally, the proportion of patients using csDMARDs (methotrexate, leflunomide, azathioprine, sulfasalazine, and mycophenolate) and glucocorticoids was similar in both groups. However, the responder group used chloroquine more frequently than the non-responder group (17% vs. 6.6%, respectively; *p* = 0.034). Non-responders had higher ACPA concentrations (156 IU/mL (60; 277 IU/mL)) than responders (46 IU/mL (6; 111 IU/mL)) (*p* < 0.001).

### 3.2. Comparison of CLP Levels between RA Responders and Non-Responders

Figure 1 compares the CLP levels between RA responders and non-responder patients. YKL-40 levels (Figure 1A) were higher in non-responders (1.82 ng/mL (1.47 ng/mL; 2.15 ng/mL)) than in responders (1.20 ng/mL (1.00 ng/mL; 1.38 ng/mL)) (*p* < 0.001). YKL-39 levels (Figure 1B) were also higher in non-responders (4.14 ng/mL (2.67 ng/mL; 6.17 ng/mL)) compared to responders (2.15 ng/mL (1.43 ng/mL; 2.73 ng/mL)) (*p* < 0.001). Furthermore, SI-CLP levels (Figure 1C) were higher in non-responders (4.65 ng/mL (2.68 ng/mL; 6.10 ng/mL)) compared to responders (2.10 ng/mL (1.11 ng/mL; 3.33 ng/mL)) (*p* < 0.001).

### 3.3. Correlations between CLP Concentrations and Selected Clinical Characteristics of Patients with RA

Positive correlations were found between the serum levels of YKL-40, YKL-39 (rho = 0.39, *p* < 0.001), and SI-CLP (rho = 0.23, *p* = 0.011), as well as between the serum levels of YKL-39 and SI-CLP (rho = 0.34, *p* < 0.001). Additionally, a positive correlation was detected between the YKL-39 protein and stiffness assessment using the visual analog scale (rho = 0.22, *p* = 0.002) and between the SI-CLP protein and the total dose of glucocorticoids (rho = 0.14, *p* = 0.040). Other statistically significant correlations between CLPs and clinical variables such as HAQ-Di, DAS28 score, erythrocyte sedimentation rate, and anti-citrullinated peptide antibodies are shown in Figure 2.

### 3.4. Quantile Regressions

Quantile regression models were performed for the 25th, 50th, and 75th percentiles of the YKL-40, YKL-39, and SI-CLP distributions. Three models were performed: the first included YKL-40 as an independent variable, the second included YKL-39 as an independent variable, and the third included SI-CLP as an independent variable. The independent variables included in all three models were patient age, disease duration, anti-CCP antibody titers, rheumatoid factor titers, and total cumulative glucocorticoid dose. Additionally, the two CLPs that were not considered dependent variables in each model were included as independent variables.

For YKL-40, it was observed that the DAS28 score was positively associated in the 0.25 and 0.75 quantiles, indicating a varied relationship depending on YKL-40 concentrations. Additionally, YKL-39 and SI-CLP showed a significant positive effect on YKL-40 in the 0.25 and 0.75 quantiles, but not in the median quantiles, suggesting a relationship between CLPs, especially at high and low values. Other variables, such as age, disease progression, anti-CCP titers, and rheumatoid factor, were not associated with YKL-40 in the quantile regression analysis.

In the case of YKL-39, quantile regression results indicated that both YKL-40 and DAS28 scores were predictors of higher levels of YKL-39 in all three quantiles evaluated. However, YKL-39 and SI-CLP were only associated with the 0.25 quantile. Conversely, the anti-CCP coefficients indicated a positive effect in all quantiles, being significant in the 0.25 and 0.75 quantiles. Variables such as age and disease duration did not show a significant impact on YKL-39 levels, suggesting that these factors are not determinants of the variability of this biomarker in patients with RA.

The quantile regression model for SI-CLP showed that for the DAS28 score, in all quantiles, there was a significant increase in SI-CLP levels associated with an increase in the DAS28 score. For YKL-39, a significant relationship was observed only in the 0.5 and 0.75 percentiles, indicating that the relationship between YKL-39 and SI-CLP is present at medium and high levels. Anti-CCP titers were associated with SI-CLP only in the 0.75 quantile. In contrast, other variables such as YKL-40, age, disease progression, and rheumatoid factor titers did not show an effect on SI-CLP.

Appendix A show the coefficients and their confidence intervals for quantile regressions of YKL-40, YKL-39, and SI-CLP, respectively.

### 3.5. ROC Curves of CLPs

Table 2 displays the AUC, 95% confidence interval, sensitivity, specificity, positive predictive value, negative predictive value, and likelihood ratio of the selected cut-off value according to the Youden index for CLPs to identify non-responder patients. The ROC curves for RA responders and non-responders showed AUC values for YKL-40, YKL-39, and SI-CLP, 0.776, 0.805, and 0.759, respectively (all *p* < 0.001). The optimal cut-off values according to the Youden index for YKL-40, YKL-39, and SI-CLP to differentiate RA responders from non-responders to csDMARDs were 1.38 ng/mL, 3.26 ng/mL, and 4.10 ng/mL, respectively. ROC curves are provided in Figure 3.

### 3.6. Logistic Regression Models

To select the reference model, the disease duration, combined DMARD therapy, leflunomide use, total dose of glucocorticoids, and anti-citrullinated peptide antibodies were evaluated as independent variables. However, in the final reference model, only HAQ-Di, ESR, ACPA titles, and total dose of glucocorticoids were included as these variables were identified as the only variables associated with treatment failure. This final model was used to compare the performances of the remaining models. Appendix A presents the results of the final model (reference model) and previous models.

### 3.7. Models for AUC Comparison Using the DeLong Test

Table 3 shows the performance evaluation of the logistic regression models 1, 2, 3, 4, and 5 to identify factors associated with treatment failure based on csDMARDs, and their comparison with the reference model. Model 2, which included variables associated with treatment failure from the reference model and the three CLPs, achieved the highest AUC and was statistically significant compared to the reference model (0.808 vs. 0.911; *p* < 0.001). Furthermore, other models that assessed the addition of CLPs individually or in combination with the reference model showed improved performance compared with the reference model (model 3, *p* = 0.019; model 4, *p* = 0.027; model 5, *p* = 0.055; model 6, *p* = 0.001; model 7, *p* < 0.001; model 8, *p* = 0.012; and model 9, *p* = 0.037). The comparison of models 6 to 15 is presented in Appendix A.

### 3.8. Chitinase Combinations (Positivity–Negativity)

Among the non-responder patients, 35.87% were positive for all three evaluated chitinases; approximately half of this percentage (17.39%) was positive for both YKL-40 and YKL-39. The third highest percentage of patients with positivity was observed in those who only tested positive for YKL-40, accounting for 14.13%. These percentages represented 18.13%, 8.79%, and 7.14% of the total patients included in this study, respectively. In the responder group, more than half of the patients did not have chitinase positivity (56.67%). Responder patients positive for YKL-40 and SI-CLP accounted for 14.44% and 11.11% of the total patients in this group, respectively. Appendix A presents the total results for both patient groups and the positivity or negativity for various chitinases as well as all possible combinations, along with Figure 4 depicting an alluvial plot.

## 4. Discussion

In the present study, the possible use of serum levels of YKL-40, YKL-39, and SI-CLP proteins as biomarkers to identify patients with csDMARD treatment failure is described for the first time. These proteins demonstrated suitable sensitivity, specificity, and predictive value for the identification of patients with RA who do not respond to csDMARDs after 6 months of use. Furthermore, the performance of regression models that included CLPs suggests that measuring CLPs is better for characterizing non-responding patients with RA to csDMARDs than not including them. Interestingly, we observed, for the first time, a positive correlation between serum levels of YKL-40 and YKL-39 and SI-CLP in patients with RA.

Of the three CLPs evaluated as biomarkers in this study, YKL-40 is the most studied in both RA and other diseases [17,39]. In this study, we observed that RA patients non-responding to csDMARDs had higher YKL-40 concentrations than responding patients. In RA, treatment failure is defined as a lack of response or limited efficacy in achieving remission or low disease activity for at least three consecutive months, requiring treatment adjustment [15,40]. Therefore, RA patients with treatment failure are characterized by a persistent inflammatory status due to high concentrations of acute-phase reactants (CRP and ESR), proinflammatory cytokines (IL-1, IL-6, TNF-α), and autoantibodies (ACPAs, RF) [10,41,42]. Therefore, YKL-40 is produced by macrophages [43], chondrocytes [44], synoviocytes [45], synovial fibroblasts [46], and neutrophils [47], and its elevation in non-responding patients may be explained by the inflammatory state. Moreover, there is evidence demonstrating a correlation between YKL-40 levels, disease activity, and persistent inflammatory status in patients with RA [19,20,48,49]. A previous study has reported elevated serum levels of YKL-40 in patients with RA with high and moderate disease activity compared to those with low disease activity and remission based on the DAS28 score [19]. Furthermore, existing information suggests that YKL-40 could have an active role in the pathogenesis of RA stimulating the production of inflammatory mediators such as chemokines (CCL2 and CXCL2), proinflammatory cytokines, and metalloproteinases (MMP-3 and MMP-9), and activating connective tissue growth factor (CTGF), favoring the formation of pannus, a determinant element in the development of erosions and joint disorganization characteristic of this disease [50,51,52,53,54]. Collectively, this evidence supports the hypothesis that YKL-40 is elevated during inflammatory states.

In our study, we also observed that elevated serum levels of YKL-40 were positively correlated with patient disability (evaluated by the HAQ-Di score), and concentrations of ACPAs that are considered classic parameters of deterioration in patients with RA [55,56,57,58,59,60]. Moreover, RA patients who do not respond to treatment often experience increased disease progression, greater loss of functionality, worse prognosis, and elevated ACPA concentrations, consistent with our findings [10,28,29,61,62,63,64,65].

In the case of YKL-39, we also observed that non-responding patients treated with csDMARDs had higher serum levels than responding patients. To our knowledge, this is the first study to correlate elevated serum levels of YKL-39 in patients with RA with treatment failure. Moreover, we found positive correlations among YKL-39, HAQ-Di, and ACPAs. The involvement of YKL-39 in diseases other than RA has been reviewed and described [22,66]. In osteoarthritis, YKL-39 is involved in the initial degeneration of cartilage and tissue remodeling [17,67]. A study observed higher serum levels of YKL-39 in patients with early-stage multiple sclerosis compared to healthy subjects [68]. Similarly, YKL-39 levels in cerebrospinal fluid were higher in patients with amyotrophic lateral sclerosis (ALS) than in controls or healthy subjects [69]. Therefore, our study strengthens the hypothesis that YKL-39 can be used as a biomarker associated with treatment failure, as there is evidence of its role in the pathogenesis of various diseases. In addition, in our study, we observed positive correlations with laboratory parameters commonly used in RA, such as ACPAs and ESR, and clinical parameters, such as HAQ-Di and DAS28-ESR, suggesting that elevated YKL-39 could be associated with persistent inflammatory status in RA patients who do not respond to treatment and their prognosis.

Regarding SI-CLP, information related to RA is very limited. However, in murine models with induced RA, it has been described that SI-CLP is attached to the surface of macrophages, increases the expression of IL-1β, IL-6, and IL-12, and worsens the inflammatory state of this disease [70]. SI-CLP is secreted by monocytic cells [70] and promotes the production of proinflammatory cytokines [70]. This information is consistent with what we found in our study, where we observed an elevation of SI-CLP in RA patients who did not respond to treatment and a correlation with some RA prognostic markers (ACPAs). The evidence thus far suggests that this relationship may be due to an increase in macrophage activity and the establishment of an inflammatory state with the help of SI-CLP, which favors the production of proinflammatory cytokines [70,71]. Another interesting contribution was the correlation between SI-CLP and the total dose of glucocorticoids. This could be explained by a cell culture study of macrophages, where it was shown that SI-CLP is regulated and stimulated by glucocorticoids by inhibiting the stabilin-1 receptor through the regulation of a lysosomal pathway [71].

Interestingly, in our study, we observed a direct relationship between the serum concentrations of these three CLPs. This positive correlation among the three CLPs can be explained by the structural relationship between YKL-40, YKL-39, and SI-CLP [16,23,24,25,72,73]. Furthermore, the structural relationship between YKL-40 and YKL-39 has been described in a gene interaction analysis performed using a web prediction tool, where the interaction between the two proteins was co-expressed [16]. In a study of patients with optic neuritis and CIS, YKL-39 levels were positively correlated with YKL-40 levels in cerebrospinal fluid [74]. Even though the Møllgaard et al. study was conducted in another autoimmune disease, to our knowledge it is the only evidence where a correlation between YKL-40 and YKL-39 has been described. The relationship between YKL-40, YKL-39, and SI-CLP described in this study, has been reported for the first time in patients and may raise new hypotheses regarding the relationship between the pathophysiology of these three proteins. However, further studies specifically designed for this purpose are necessary.

The sensitivity and specificity values obtained for CLPs were generally good [75]. For the YKL-40 protein, the sensitivity and specificity values (sensitivity: 80.43%, specificity: 74.44%) indicated its ability to detect 76.04% of patients who did not respond to csDMARDs. However, the LR values (LR+ (3.15) and LR− (0.26)) are not optimal [75,76,77]; therefore, the clinical utility of evaluating this marker should be performed with other types of studies that can measure changes in YKL-40 and control other factors related to the elevation of this protein.

Regarding the clinical utility results of the YKL-39 and SI-CLP proteins, the sensitivity and specificity values of the cutoff point for both proteins indicate that they have a good capacity [75] to detect patients with and without treatment failure. However, the estimation of LR+ and LR− indicates that more studies are needed to understand the clinical value of measuring these proteins in clinical practice.

In this study, in addition to evaluating sensitivity, specificity, predictive value, and likelihood ratio, the utility of CLPs as potential biomarkers for treatment failure was assessed using logistic regression models. Since we did not have other biomarkers, variables commonly used in RA patients (HAQ-Di, ESR, total dose of glucocorticoids, and ACPAs) were used as a reference model. Compared with this model, the measurement of CLPs adds diagnostic precision, and improves the accuracy of the model, increasing the AUC from 0.808 to 0.911 (DeLong test; *p* < 0.001) [78] and could lead to potential clinical benefits in patients with csDMARDs, such as the possibility of early and personalized interventions. This improvement could translate into a better diagnosis and prognosis for patients with RA, allowing a more precise identification of those likely to not respond to standard treatment. Additionally, this improvement in discrimination could lead to a reduction in the administration of ineffective treatments, enhance the quality of life of the patient, and reduce healthcare costs. The results of the models, as well as the positive correlations between the three CLPs evaluated in this research, suggest that the three CLPs may have an impact on RA patients not responding to csDMARDs. Furthermore, these results suggest that the joint measurement of the three proteins improves performance more than the individual addition to the models.

This study has several limitations. First, this study does not prove that the elevation of CLPs is a phenomenon exclusive to patients with DMARD failure. Therefore, it is necessary to generate more information to determine which other biological and pathological conditions elevate CLPS. Another limitation of our study is its design and temporality. Our study was not designed to evaluate the changes in CLP concentrations in patients with RA. Therefore, we could not determine whether the increase in CLPs occurred before or after treatment failure. To validate the clinical usefulness of these biomarkers, it is necessary to carry out cohort studies. Therefore, new studies are required to understand the temporal relationship between CLPs and treatment failure. Furthermore, although our results on sensitivity, specificity, and predictive values suggest that measuring CLPs in patients with RA could have clinical utility, the values of LR+ and LR− are not optimal. We believe that other variables may influence the concentrations of CLPs. These suboptimal LR values are also reflected in elevated CLPs in some patients responding to csDMARDs. Additionally, in this study, we did not measure pro-inflammatory cytokines or other markers of inflammation other than ESR; therefore, we cannot assert that the elevation of CLPs is due to persistent inflammation in non-responders. The validation of our hypothesis related to inflammation and the elevation of CLPs requires new studies that ideally should be proposed with other longitudinal epidemiological designs, the measurement of other variables, the measurement of other variables such as renal function and other clinical variables, including patients who begin treatment with csDMARDs as well as the inclusion of patients with biological DMARDs. In this way, it will be possible to understand the mechanisms underlying the elevation of CLPs in patients with RA. In this war, another limitation of this study is that we did not include a comparison group of patients receiving biological treatment. Therefore, we do not know whether the elevation of CLPs is observed only in patients undergoing treatment with csDMARDs or if treatment with biological DMARDs could potentially alter it. In this study, although we statistically demonstrated the direct relationship between the levels of YKL-40, YKL-39, and SI-CLP, our results do not allow us to determine whether this statistical relationship is biologically significant. Further studies are needed to evaluate the cellular mechanisms that could explain the elevation of these three CLPs.

This study had several strengths. Evaluation of CLPs as biomarkers was performed using different methods (ROC curves, logistic regressions, sensitivity, specificity, PPV, NPV, and LR). This offers greater diagnostic precision because the combined use of these indicators provides a more comprehensive and accurate view of the performance of CLPs in identifying non-responders to csDMARDs, as each of these measures provides a unique perspective on how CLPs behave in different situations. The use of this approach could provide guidelines for subsequent studies to improve therapeutic strategies. Additionally, the measurement of CLPs could be easily implemented in most laboratories. In this study, we used quantile regression models to identify the factors associated with CLP elevation. This approach allows us to understand the relationship between variables and CLPs at different points in the distribution, thereby providing a more detailed and comprehensive view of the relationship between the variables. Another strength is that the evaluation of CLPs was conducted using a multivariate approach utilizing logistic regression models, which allowed for the adjustment of confounding effects to obtain a clearer and more direct relationship between CLPs and response to treatment. Additionally, multivariate analysis allows for a more comprehensive understanding of how other variables might influence treatment failure and the elevation of CLP concentrations. Lastly, we consider another strength of this study to be our observation of correlations between YKL-40, YKL-39, and SI-CLP with clinical and laboratory parameters commonly used in patients with RA, which allows for a comparison of CLPs with variables commonly used in RA.

## 5. Conclusions

The measurement of CLPs can be a useful biomarker for identifying patients who do not respond to csDMARDs, as the inclusion of CLPs also enhances the performance of models that only incorporate clinical variables and laboratory results. However, further studies with a different epidemiological approach are required to validate these findings.

## Figures and Tables

**Figure 1 biomedicines-12-01406-f001:**
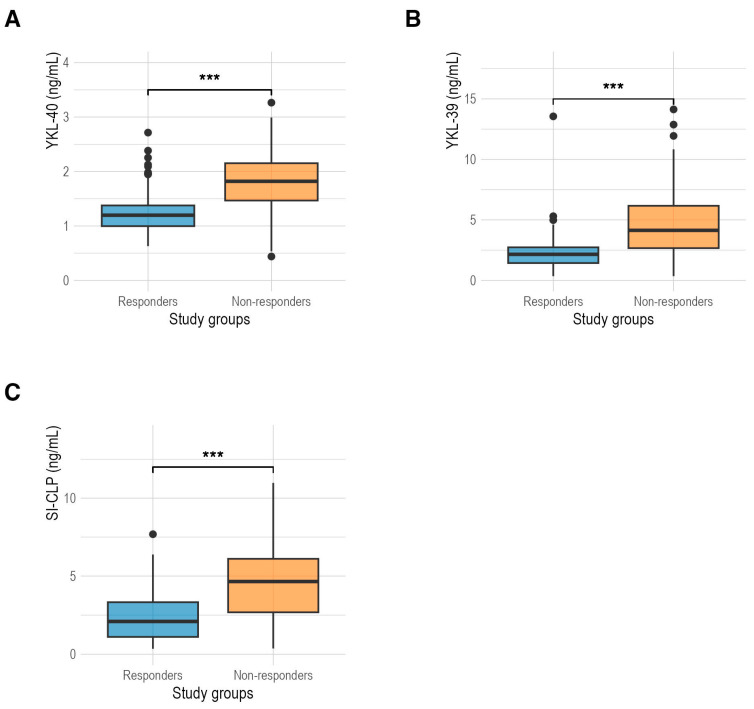
(**A**) Comparison of serum levels of YKL-40 protein between responders (1.20 ng/mL) and non-responders (1.82 ng/mL) in RA patients; (**B**) comparison of serum levels of YKL-39 protein between responders (2.15 ng/mL) and non-responders (4.14 ng/mL) in RA patients; (**C**) comparison of serum levels of SI-CLP protein between responders (2.10 ng/mL) and non-responders (4.65 ng/mL). ***: *p* < 0.001.

**Figure 2 biomedicines-12-01406-f002:**
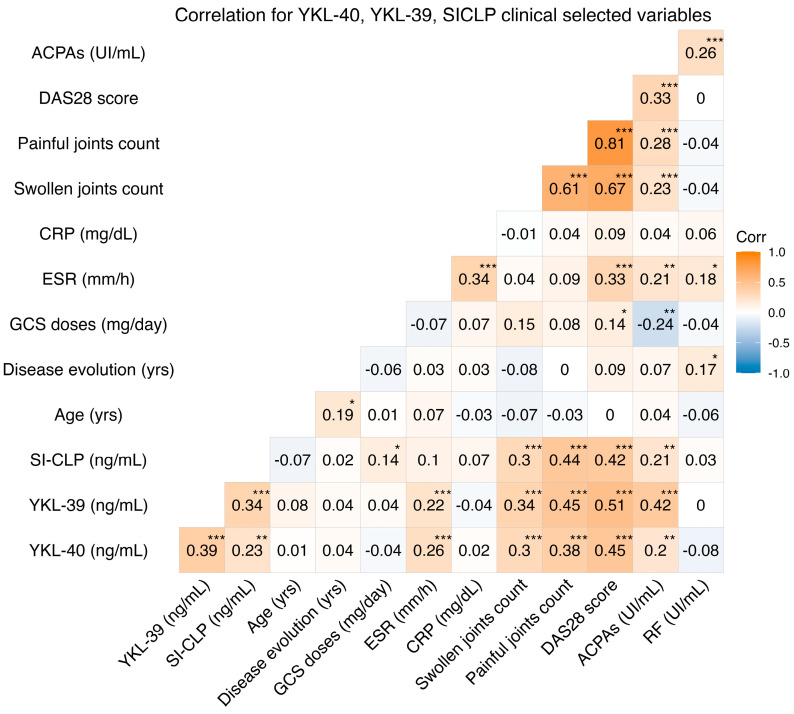
Correlations between YKL-40, YKL-39, SI-CLP, clinical variables, and laboratory values. ACPAs: anticyclic citrullinated protein antibodies; DAS28: disease activity score 28; CRP: C reactive protein; ESR: erythrocyte sedimentation rate; GCS: glucocorticoids; RF: rheumatoid factor. *: *p* < 0.05; **: *p* < 0.01; ***: *p* < 0.001.

**Figure 3 biomedicines-12-01406-f003:**
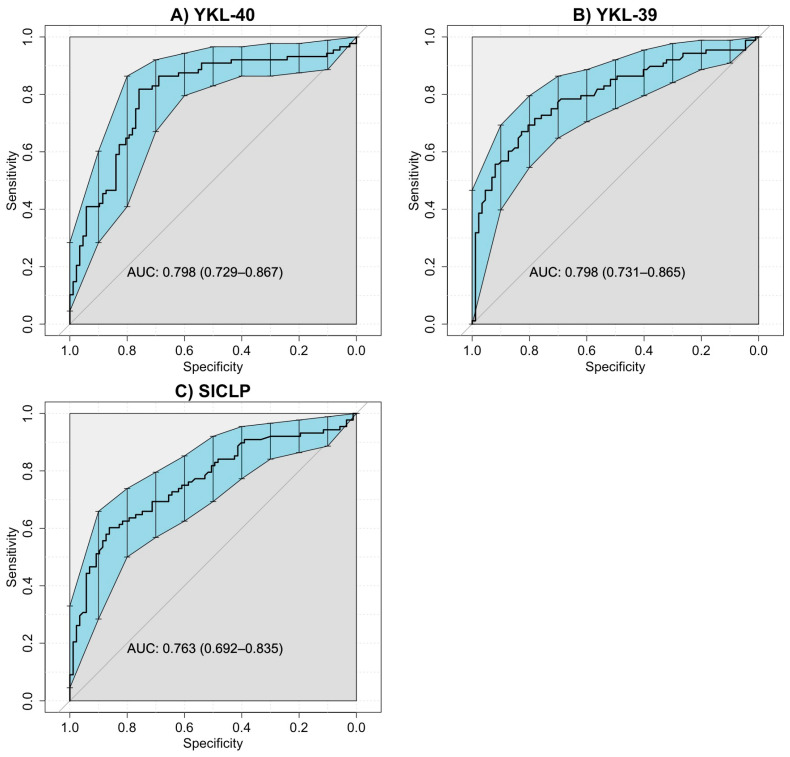
ROC curves of CLPs for RA patients included in both study groups. (**A**) ROC curve for serum levels of YKL-40 in RA patients responders and non-responders (AUC = 0.798, IC 95% 0.729–0.867). (**B**) ROC curve for serum levels YKL-39 in RA patients responders and non-responders (AUC = 0.798, IC 95% 0.731–0.865). (**C**). ROC curve for serum levels SI-CLP in RA patients responders and non-responders (AUC = 0.763, IC 95% 0.692–0.835). The blue area represents the confidence intervals for the ROC curve.

**Figure 4 biomedicines-12-01406-f004:**
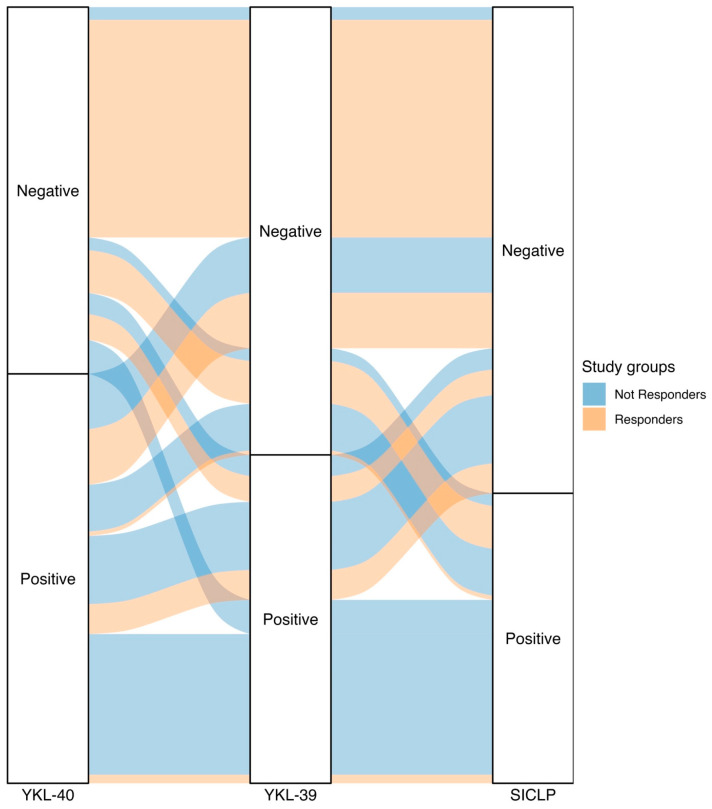
The alluvial plot depicting the classification of patients from the study groups (responders and non-responders) into positive or negative for chitinase-like proteins. Patients with RA who respond to treatment are represented in pink, and patients with RA who do not respond to treatment are represented in blue. We observe that one-third of the patients with RA who do not respond to treatment (35.87%) have elevated levels of all 3 CLPs. Additionally, more than half of the patients with RA who respond to treatment (56.67%) do not have elevated levels of all 3 CLPs.

**Table 1 biomedicines-12-01406-t001:** Comparison between RA patients with failure to DMARDs (non-responders) and RA patients with response to DMARDs (responders).

Characteristic	Responders ^1^ (DAS28 < 3.2) n = 87	Non-Responders ^1^ (DAS28 ≥ 3.2) n = 88	*p*-Value ^2^
Female, n (%)	87 (100)	88 (100)	1.00
Age, years	59 (51; 67)	59 (52; 64)	0.61
Tobacco use, n (%)	11 (13)	9 (10)	0.62
Alcohol use, n (%)	11 (13)	15 (17)	0.41
Sedentary, n (%)	54 (58)	55 (60)	0.71
Menopause, n (%)	73 (84)	77 (88)	0.50
Comorbidities			
Arterial hypertension, n (%)	40 (46)	35 (40)	0.41
Type 2 diabetes mellitus, n (%)	10 (11)	12 (14)	0.67
Disease evolution, years	11 (7; 19)	12 (8; 19)	0.60
csDMARDs, n (%)	87 (100)	88 (100)	1.00
Monotherapy, n (%)	51 (59)	44 (50)	0.25
Methotrexate use, n (%)	56 (64)	49 (56)	0.24
Methotrexate weekly doses, mg	10.0 (0.0; 15.0)	8.8 (0.0; 15.0)	0.70
Leflunomide use, n (%)	30 (34)	42 (48)	0.075
Azathioprine use, n (%)	11 (13)	14 (16)	0.54
Sulfasalazine use, n (%)	32 (37)	35 (40)	0.68
Chloroquine use, n (%)	14 (16)	5 (6)	0.029
Combined therapy, n (%)	36 (41)	44 (50)	0.25
Two csDMARDs, n (%)	33 (38)	37 (40)	0.46
More than two csDMARDs, n (%)	6 (7)	10 (11)	0.46
Other DMARDs, n (%)	9 (10)	12 (13)	0.61
Glucocorticoid use, n (%)	54 (62)	63 (72)	0.18
Glucocorticoid doses, mg/day	5 (0.00; 5.00)	5 (0.00; 6.00)	0.073
ESR, mm/h	18 (15; 29)	24 (18; 37)	<0.001
CRP, mg/dL	5 (0; 7)	5 (0; 10)	0.45
RF, UI/mL	58 (31; 95)	56 (27; 93)	0.78
ACPAs, UI/mL	52 (6; 114)	157 (60; 281)	<0.001
HAQ-DI	0.25 (0.00; 0.61)	0.88 (0.33; 1.41)	<0.001
DAS28 (previous)	2.87 (2.42; 3.60)	3.49 (2.72; 4.10)	0.002
DAS28	2.45 (2.04; 2.79)	4.32 (3.80; 5.07)	<0.001

^1^ n (%) for qualitative variables; medians and (p25; p75) for quantitative variables. ^2^ Fisher’s exact test; Mann–Whitney U test; Pearson’s chi-squared test. RA: rheumatoid arthritis, DMARDs: disease-modifying antirheumatic drugs, ESR: erythrocyte sedimentation rate, CRP: C-reactive protein, RF: rheumatoid factor, ACPAs: anti-citrullinated peptide antibodies, HAQ-DI: health assessment questionnaire disability index, DAS28: Disease Activity Score 28 combined with erythrocyte sedimentation rate.

**Table 2 biomedicines-12-01406-t002:** Responders and non-responders values of YKL-40, YKL-39, and SI-CLP in RA patients.

Parameters	AUC	95% CI	Optimal Cut-Off Value	Specificity (%)	Sensitivity (%)	PPV (%)	NPV (%)	LR+	LR−	AC (%)
YKL-40 (ng/mL)	0.798	0.729–0.867	1.392	75.86	81.82	77.42	80.49	3.39	0.24	78.86
YKL-39 (ng/mL)	0.798	0.731–0.865	3.257	82.76	67.05	79.73	71.29	3.89	0.40	74.86
SI-CLP (ng/mL)	0.763	0.692–0.835	4.063	86.21	60.23	81.54	68.18	4.37	0.46	73.14

AUC: area under curve, 95% CI: 95% confidence interval, PPV: positive predictive value, NPV: negative predictive value, LR+: positive likelihood ratio, LR−: negative likelihood ratio, AC: accuracy, YKL-40 (CHI3L1: chitinase-3-like-1), YKL-39 (CHI3L2: chitinase-3-like-2), SI-CLP: stabilin-interacting chitinase-like protein (CHID1: chitinase domain-containing protein 1).

**Table 3 biomedicines-12-01406-t003:** Selected models. Models 1 to 5 of the variables associated with treatment failure and CLPs and their comparison using DeLong test.

Models	AUC	DeLong Test
Models of variables associated with treatment failure and chitinase-like proteins		
Model 1: Variables associated with treatment failure excluding the 3 chitinase-like proteins (HAQ-Di, ESR, Total dose of glucocorticoids, ACPAs).	0.806	Not applicable
Model 2: Variables associated with treatment failure including the 3 chitinase-like proteins.	0.918	<0.001
Model 1: Variables associated with treatment failure excluding the 3 chitinase-like proteins.	0.806	
Model 3: Variables associated with treatment failure and the YKL-40 protein.	0.882	0.007
Model 1: Variables associated with treatment failure excluding the 3 chitinase-like proteins.	0.806	
Model 4: Variables associated with treatment failure and the YKL-39 protein.	0.856	0.033
Model 1: Variables associated with treatment failure excluding the 3 chitinase-like proteins.	0.806	
Model 5: Variables associated with treatment failure and the SI-CLP protein.	0.853	0.052
Model 1: Variables associated with treatment failure excluding the 3 chitinase-like proteins.	0.806	

## Data Availability

The data presented in this study will be available upon request to the corresponding author. The data are not publicly available due to privacy issues.

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
