# Peer review of "Elevated Serum Levels of YKL-40, YKL-39, and SI-CLP in Patients with Treatment Failure to DMARDs in Patients with Rheumatoid Arthritis"

_biomedicines, 2024, doi:10.3390/biomedicines12071406_

Round 1
Reviewer 1 Report (Previous Reviewer 1)
Comments and Suggestions for Authors
It seems that my opinion is not understood by the authors, so I will state it again.
There are no major problems with the contents of this paper.
However, the representation is excessive.
I understand that the authors are describing that this is the result of a study on csDMARD because they investigated csDMARD user. However, it is unclear whether this is a characteristic result found only in csDMARDs user. These are the results of a survey of csDMARD users, and it cannot be determined whether these results are characteristics of patients with csDMARDS. Therefore, I would like the author to write in a way that does not mislead the reader.
I would like you to consider this objectively for once. Would not you feel strange if the title looked like this? "Elevated serum levels of IL-6 is a potential biomarker of treatment failure with synthetic DMARDs in patients with Rheumatoid Arthritis."
RA patients with treatment failure for csDMARDs have high IL-6, but in 2024 we know that IL-6 elevation is not a phenomenon limited to treatment of csDMARDs.
It has not been proven whether an increase in chitinase-like proteins is a phenomenon limited to csDMARDs.
Would you please use expressions that are not misleading. Especially, the title.
P.S. "we consider that to understand the characteristics of a group of patients, in this case, those on csDMARDs, it is not necessary to compare them with patients on other treatments."
I completely disagree with this statement. I am very small. This small height is my characteristics. But, this "small" means "smaller than standard" or "smaller than someone". To say characteristics require comparisons.
The increase in chitinase-like proteins might be much higher in patients with biologics failure. If so, the characteristic of csDMARD failure is "does not increase much." 
Have you ever considered such a possibility?
Author Response
Comment 1
“However, the representation is excessive.
I understand that the authors are describing that this is the result of a study on csDMARD because they investigated csDMARD user. However, it is unclear whether this is a characteristic result found only in csDMARDs user. These are the results of a survey of csDMARD users, and it cannot be determined whether these results are characteristics of patients with csDMARDS. Therefore, I would like the author to write in a way that does not mislead the reader.
I would like you to consider this objectively for once. Would not you feel strange if the title looked like this? "Elevated serum levels of IL-6 is a potential biomarker of treatment failure with synthetic DMARDs in patients with Rheumatoid Arthritis."
RA patients with treatment failure for csDMARDs have high IL-6, but in 2024 we know that IL-6 elevation is not a phenomenon limited to treatment of csDMARDs.
It has not been proven whether an increase in chitinase-like proteins is a phenomenon limited to csDMARDs.
Would you please use expressions that are not misleading. Especially, the title.”
Answer
Thank you for your comments. We would like to clarify this point. Our intention with the title was never to imply that CLPs are biomarkers for csDMARD failure. We completely agree with your opinion on being objective in our work. We have decided to make changes to the title to allow readers to understand that CLPs are not a validated biomarker for failure. However, in the future, with different epidemiological designs, other comparison groups (such as patients on biologics), and after obtaining more information, they could potentially be a good biomarker.
The title changed and now is: "Elevated serum levels of YKL-40, YKL-39, and SI-CLP in patients with treatment failure to DMARDs in patients with Rheumatoid Arthritis", therefore, we believe it is less presumptuous.
Besides, we added the following sentences in the discussion section:
This study has several limitations. First, this study does not prove that the elevation of CLPs is a phenomenon exclusive to patients with DMARDs failure. Therefore, it is necessary to generate more information to determine which other biological and pathological conditions elevate CLPS. (lines 411-414).
Comment 2
P.S. "we consider that to understand the characteristics of a group of patients, in this case, those on csDMARDs, it is not necessary to compare them with patients on other treatments."
I completely disagree with this statement. I am very small. This small height is my characteristics. But, this "small" means "smaller than standard" or "smaller than someone". To say characteristics require comparisons.
The increase in chitinase-like proteins might be much higher in patients with biologics failure. If so, the characteristic of csDMARD failure is "does not increase much." 
Have you ever considered such a possibility?
Answer
We agree with your comment. Unfortunately, we do not have the necessary data or samples to perform the comparisons you suggest, which we also agree are imperative to lay the foundation for the possible validation of CLPs as biomarkers. In the text, we have listed the limitations of this study, highlighting the temporality and types of treatments used by the patients as the most significant and the first to be modified in a prospective cohort study that includes patients on biologics and synthetics. With this, we aim to incorporate your appropriate perspective on the potential and limitations of this study.
Thank you for helping us to improve the quality of this work.
Reviewer 2 Report (Previous Reviewer 3)
Comments and Suggestions for Authors
All my comments were addressed.
Author Response
Thank you for your time to the revision of this work.
This manuscript is a resubmission of an earlier submission. The following is a list of the peer review reports and author responses from that submission.
Round 1
Reviewer 1 Report
Comments and Suggestions for Authors
It has been reported that serum chitinase-like protein YKL-40 is high in patients with severe rheumatoid arthritis. (For example; Matsumoto T, et al.Clin Exp Rheumatol. 2001 Nov-Dec;19(6):655-60.) Furthermore, there are multiple reports that YKL-40 correlates with IL-6 in serum. (For example;Brahe CH, et al. Scand J Rheumatol. 2018 Jul;47(4):259-269.) There is a report that YKL-40 is high in PsA where CRP is elevated, but there is a report that YKL-39 is increased but YKL-40 is not increased in osteoarthritis where IL-6 is not high. (Knorr T, et al. Ann Rheum Dis. 2003 Oct;62(10):995-8.) In this study, the authors report that YKL-40 and YKL-39 levels are high in patients with treatment failure for csDMARDs. I think these findings are interesting, but I also think their conclusions are very self-righteous.
First of all, why were the authors able to conclude that these molecules are elevated in patients with csDMARD treatment failure? Are there any differences between patients with treatment failure on csDMARDs compared to patients with treatment failure on Biologics or JAK inhibitors?
Also, is it necessary to be a patient with treatment failure? 
Shouldn't it be correlated with the disease activity of rheumatoid arthritis? 
Did the administration of csDMARS affect the results of this study?
The authors write in the introduction that csDMARD is used regulary in Latin America.
Most of the patients with treatment failure for rheumatoid arthritis might be treatment failure patients for csDMARDs. However, the characteristics of patients treated with csDMARDs cannot be determined without comparison with data from patients treated with other treatments. The authors have written "csDMARD" many times, but the author's study design is incorrect for writing this term. The authors have repeatedly written about treatment failure. A before-and-after comparison is necessary to show that this is a treatment failure rather than the disease activity of the first visit patient. The author's study design is incorrect for writing this term.
Minor point
There is no period on line 336.
Reviewer 2 Report
Comments and Suggestions for Authorsž Biological DMARDs are very important for treating RA these days. Why did the authors exclude?
ž DAS28-ESR at basline (before treatment) should be provided.
ž Effect of methotrexate varies significantly depending on doses. Please analyze again considering prescribed doses.
ž Renal function affects significantly selection of DMARDs and those doses. Please analyze again considering renal fucntion.
ž Table 1: ‘Monotherapy ’ should be described in detail.
ž How about details in combination therapy of DMARDs in the present study?
Reviewer 3 Report
Comments and Suggestions for Authors
The article presents an important topic for the management of patients with rheumatoid arthritis.
I have minor comments.
Study design
- Include a sentence stating the case-control study.
Results
- How was the sedentary status assessed?
Discussion
- Are the specific chitinase-like proteins affordable for all the patients? Have all the laboratory the capacity to do these tests?
Reviewer 4 Report
Comments and Suggestions for Authors
1. Please use in describing significant p-value: p < 0.05, p < 0.01 and p < 0.001 according to the raw value.
2. Please provide the sample size calculation with proper assumptions.
3. Please use the same brackets for a description of the lower and upper quartiles both in tables and text, I suggest: median (lower quartile; upper quartile).
4. Showing the mutual correlations in Figure 2 should be followed with a proper analysis of factors related to YKL-40, YKL-39, and SI-CLP levels such as quantile regression or other regression techniques.
5. It does not make any sense to do and parallel analyze so many ROC models. Instead of this authors should either present the one, maybe two best ones. I propose to use f.e. ROC regression to provide the best model based on AIC or BIC criteria.
6. The main issue with the results is, that they do not present the cohort study results, so there are no measurements at the beginning, during, and at the end of the study. Thus no biomarkers longitudinal analysis can be done.
7. As there are only 7 men in the group I propose to remove them from the analysis, recalculate all analysis (taking the mentioned hints), and change the title to "... women with rheumatoid arthritis".
8. The following factors should be added: time of the disease,
